# Howland Forest, ME, USA: Multi-Gas Flux (CO$_2$, CH$_4$, N$_2$O) Social Cost Product Underscores Limited Carbon Proxies

**Bruno D. V. Marino \*** , **Nahuel Bautista and Brandt Rousseaux**

Executive Management, Planetary Emissions Management Inc., Cambridge, MA 02139, USA;
nahuel.bautista@pem-carbon.com (N.B.); brandt.rousseaux@pem-carbon.com (B.R.)
\* Correspondence: bruno.marino@pem-carbon.com

**Abstract:** Forest carbon sequestration is a widely accepted natural climate solution. However, methods to determine net carbon offsets are based on commercial carbon proxies or CO$_2$ eddy covariance research with limited methodological comparisons. Non-CO$_2$ greenhouse gases (GHG) (e.g., CH$_4$, N$_2$O) receive less attention in the context of forests, in part, due to carbon denominated proxies and to the cost for three-gas eddy covariance platforms. Here we describe and analyze results for direct measurement of CO$_2$, CH$_4$, and N$_2$O by eddy covariance and forest carbon estimation protocols at the Howland Forest, ME, the only site where these methods overlap. Limitations of proxy-based protocols, including the exclusion of sink terms for non-CO$_2$ GHGs, applied to the Howland project preclude multi-gas forest products. In contrast, commercial products based on direct measurement are established by applying molecule-specific social cost factors to emission reductions creating a new forest offset (GHG-SCF), integrating multiple gases into a single value of merit for forest management of global warming. Estimated annual revenue for GHG-SCF products, applicable to the realization of a Green New Deal, range from ~\$120,000 USD covering the site area of ~557 acres in 2021 to ~\$12,000,000 USD for extrapolation to 40,000 acres in 2040, assuming a 3% discount rate. In contrast, California Air Resources Board compliance carbon offsets determined by the Climate Action Reserve protocol show annual errors of up to 2256% relative to eddy covariance data from two adjacent towers across the project area. Incomplete carbon accounting, offset over-crediting and inadequate independent offset verification are consistent with error results. The GHG-SCF product contributes innovative science-to-commerce applications incentivizing restoration and conservation of forests worldwide to assist in the management of global warming.

**Keywords:** California Air Resources Board; Climate Action Reserve; eddy covariance; Howland Forest; social cost of CO$_2$; CH$_4$; N$_2$O

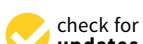

## 1. Introduction

Uncertainty and high cost of typical commercial forest carbon offset protocols are unresolved [1–6], impeding widespread adoption and expansion of forest conservation projects. The main endeavor of commercial forest carbon offset trading is to assist landowners with the conservation and restoration of forests based on the net carbon sequestration and carbon credit sales for a project [7,8] while verifiably reducing net emissions. While forest restoration is recognized as a viable, economic, and readily deployable nature-based commercial solution to mitigate climate change [9–14], forest loss continues at a rate of ~10 million hectares annually from 2015–2020 [15], outpacing restorative efforts. In contrast, the forest landscape conserved by carbon protocols and trading is astonishingly small, ~0.03% of the available land for restoration of ~0.9 billion hectares [12,15], evidence that existing methods underpinning forest carbon are not economically or ecologically viable. Forest carbon sequestration credits, typically derived from sparse forest mensuration (6- or 12-year timber inventory) [16–18] surveys for above-ground carbon and use of multiple, carbon denominated growth models [18–20], by default, exclude direct measurement of GHG's, limiting innovative commercial applications.

In contrast, direct, hourly, in situ measurement of forest greenhouse gas (GHG) fluxes, via eddy covariance and soil accumulation chambers are widely used and accepted forest research methods [21,22]. Such methods result in net ecosystem exchange (NEE) [22–24], integrating vertical gross fluxes between the forest, soils, and the atmosphere [25–29]. The NEE approach, reported in 600+ forest carbon and GHG studies [30], provides the foundation for commercial applications across small and large landscapes [1,31,32], and three-gas forest products ($CO_2$, $CH_4$, $N_2O$), integrating gas flux sources (tower and soil accumulation chambers) with pricing guidelines to establish commercial products. Prices for $tCH_4$ and $tN_2O$ emission reductions and abatement costs are not well established [7]. However, the social cost of GHGs represents, in theory, the monetary value of the net harm to society linked to the emission of one ton of a GHG each year and is used here to value social cost linked offsets [33]. Emissions linked to their social cost and corresponding offset revenue potential, now and in the future, are fundamental to the realization of Green New Deal policies [34] mandating emission reductions, but lacking specificity of the role of forests and GHG pricing in green policies [35–37]. The expense and operational challenges of measuring multiple GHGs are justified given their high global warming potential (e.g., $CH_4$ (28–36x); $N_2O$ (265–298x), 100-year lifetime) relative to $CO_2$ [33], the availability of ~4 billion forested hectares globally, the high value of GHG data for fundamental ecosystem studies and model validation [38], and as innovative GHG forest products.

In this study, we analyze the Howland Forest GHG flux measurement record (tower and soil accumulation chamber) to determine the net multi-gas budget for the period studied. The GHG data are monetized using social cost pricing factors for each gas, demonstrating a new GHG forest product (GHG-SCF). We then analyze annual NEE reported as $gC\ m^{-2}\ y^{-1}$, or equivalent units, for two adjacent Howland Forest $CO_2$ flux towers, quantifying differences between the California Air Resources Board [16] and the Climate Action Reserve (CARB-CAR) [39] data [40,41]; Howland is the only site where both methods have been used contemporaneously. The CARB-CAR third-party validation process, a critical link in the carbon offset supply chain, is evaluated for error sources and non-adherence to regulatory provisions, including assessment of independent verification methods for CARB-CAR forest carbon offsets. We conclude by suggesting that GHG-SCF projects should be undertaken to establish holistic net GHG budgets applicable to global forest management while providing enhanced revenue for landowners.

## 2. Materials and Methods

### 2.1. Site Description

The Howland Forest (Figure 1) is located in central Maine at about 5 km southwest of the Howland town and 56 km north of Bangor (45.2041° N 68.7402° W, elevation 60 m above sea level). It covers 557 acres (~225 ha) and is classified as an evergreen needleleaf forest (ENF; lands dominated by woody vegetation with a percent cover >60% and height exceeding 2 m. Almost all trees remain green all year) according to the International Geosphere-Biosphere Program (IGBP). The stands are about 20 m tall and consist of spruce-hemlock–fir, aspen–birch, and hemlock–hardwood mixtures, which were logged selectively around 1900. More information can be found on its website https://umaine.edu/howlandforest/ (accessed on 1 March 2021).

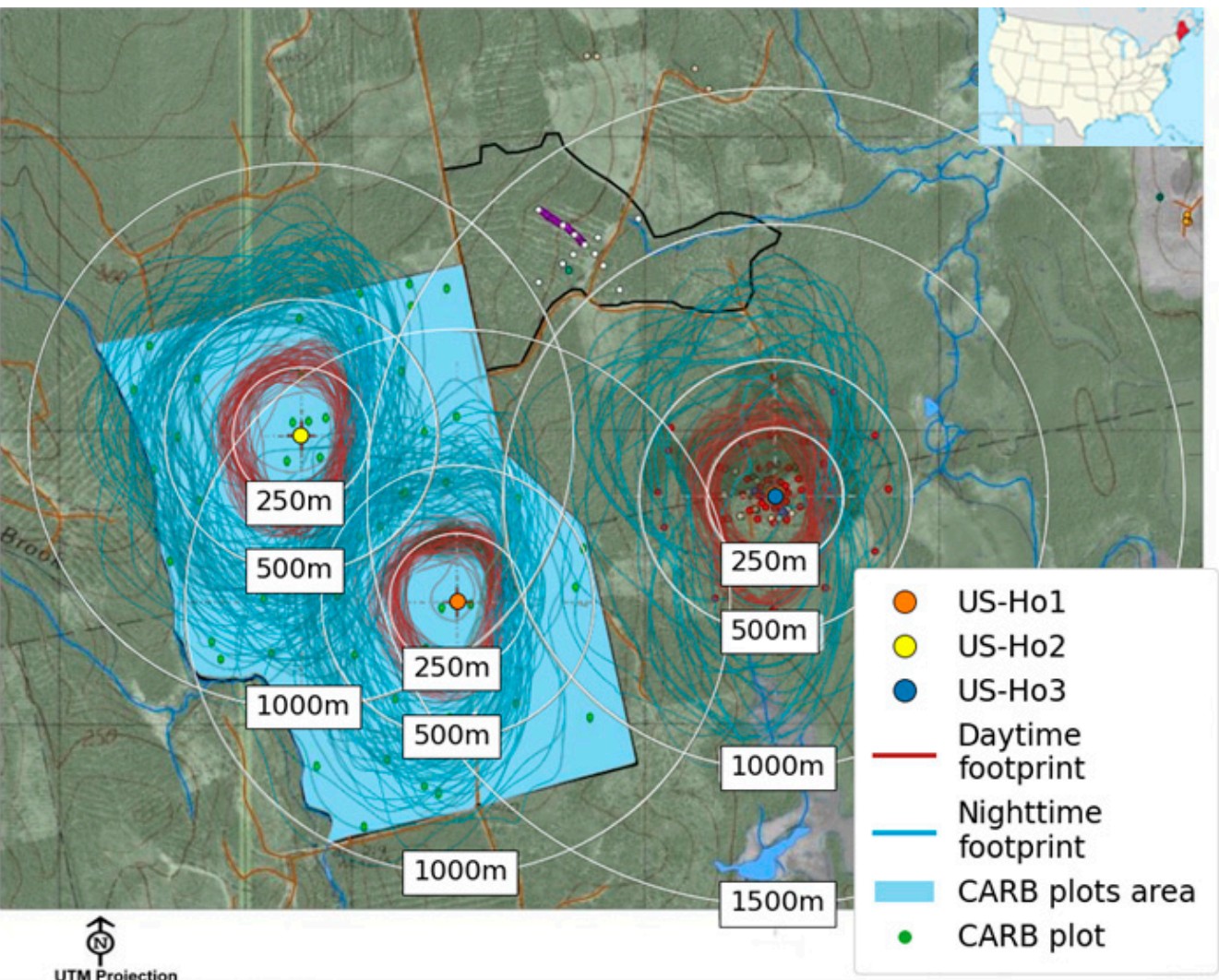

**Figure 1.** Map of the Howland research area. Tower locations (yellow circles; the towers are: Ho2 west, Ho1 center, and Ho3 east), daytime and nighttime monthly climatology footprints (red and light-blue lines, respectively), California Air Resources Board CARB) measurements area (light blue area), and CARB individual plots (filled green circles) are represented. As a reference, white circles indicate the areas within 250, 500, 1000, and 1500 m around each tower. Small white, purple, red, and orange dots belong to the original base map and correspond to measurements done at the Howland site, but they are not relevant to this study. The map was redrawn from https://umaine.edu/howlandforest/about/ (accessed on 25 March 2021) and [17]; monthly climatology data are from [42]. Note: $CO_2$ fluxes used in this study were measured above the canopy at a 29 m tower with the eddy covariance technique since 1996 (US-Ho1; "the main tower"), from 1999 to 2004 (US-Ho2; "west tower"), and from 2004 to 2007 (US-Ho3). US-Ho1 includes $CH_4$ measurements from 2012 to 2018, and it is approximately 775 m apart 116 from US-Ho2.US map insert acknowledgment:https://commons.wikimedia.org/wiki/File: Maine_in_United_States.svg (accessed on 2 March 2021).

### 2.2. $CO_2$ and $CH_4$ Tower Fluxes

Howland has the second longest-running flux record in the United States, dating back to 1996 (the longest belonging to Harvard Forest). These 20 years of data provide a time-series long enough for robust analyses of relationships between NEE and various environmental variables. $CO_2$ fluxes used in this study were measured above the canopy at a 29 m tower with the eddy covariance technique since 1996 (US-Ho1; "the main tower"), from 1999 to 2004 (US-Ho2; "west tower"), and from 2004 to 2007 (US-Ho3). US-Ho1 includes $CH_4$ measurements from 2012 to 2018, and it is approximately 775 m apart from US-Ho2. The additional tower, US-Ho3, was used to monitor NEE after a

shelterwood harvest [43]. Removal of biomass from the project area was negligible for the areas represented by US-Ho1,2, while US-Ho3 experienced the planned shelterwood harvest to record changes in NEE [43]. More in-depth details about flux and footprint measurements and preprocessing can be found at [38,40,41]. Preprocessed data before filtering and gap-filling can be found at the AmeriFlux website (https://ameriflux.lbl.gov/sites/site-search/#keyword=Howland (accessed on 27 March 2021) or in each tower repository:

Us-Ho1: David Hollinger (1996-) AmeriFlux US-Ho1 Howland Forest (main tower), Dataset. https://doi.org/10.17190/AMF/1246061 (accessed on 27 March 2021).

US-Ho2: David Hollinger (1999-) AmeriFlux US-Ho2 Howland Forest (west tower), Dataset. https://doi.org/10.17190/AMF/1246062 (accessed on 27 March 2021).

US-Ho3: David Hollinger (2000-) AmeriFlux US-Ho3 Howland Forest (harvest site), Dataset. https://doi.org/10.17190/AMF/1246063 (accessed on 27 March 2021).

### 2.3. $CO_2$, $CH_4$, and $N_2O$ Soil Fluxes

An automated chamber system was used to measure soil $CO_2$, $CH_4$, and $N_2O$ fluxes within the footprint of the US-Ho1 tower from 2012 to 2016, approximately once per hour during the snow-free period when vegetation was active (from May to November). Exact locations where the chambers were installed varied among years. Each chamber was 30.5 cm in diameter. Between measurements, the chamber top was lifted using a pneumatic piston off a PVC collar permanently inserted into the soil surface [38]. More details can be found at [44,45]. The data can be downloaded from [38].

### 2.4. Data Processing and Calculations

$CO_2$ eddy covariance data were processed with REddyProc 1.2.1 [46], which filters low turbulence data using the methodology from [47] (with the 50-percentile criterion) and then fills all the gaps produced by the filtering technique or by instrument failure with a lookup table. The soil temperature at the lowest depth was chosen as the input variable for REddyProc along with the above canopy air temperature ($T_{air}$), the vapor pressure deficit and the photosynthetic photon flux density divided by 0.47 as global radiation.

Ecosystem respiration ($R_{eco}$), its photosynthesis (gross primary productivity; *GPP*) and *NEE* are related according to the equation:

$$NEE = R_{eco} - GPP \tag{1}$$

In this study, $R_{eco}$ was estimated with REddyProc based on the nighttime approach [21,22], which fits the Lloyd and Taylor [48] model for respiration (Equation (2)) using only night-time data, because $NEE = R_{eco}$ at night, and then extrapolating the parameters $R_{ref}$ and $E_0$ found in the regression to calculate daytime $R_{eco}$ ($T_{ref}$ and $T_0$ are fixed). Then, *GPP* is calculated with Equation (1) [49].

$$R_{eco} = R_{ref} \, e^{\frac{E0}{T_{ref} - T0} - \frac{E0}{T_{air} - T0}} \tag{2}$$

Afterward, yearly *NEE*, $R_{eco}$ and *GPP* sums were calculated in Python 3.7.7. In the literature, *NEE* can also be expressed as net ecosystem production (*NEP*), where *NEP* = *NEE* [50].

### 2.5. GHG Forest and Social Cost of $CO_2$, $CH_4$ and $N_2O$

Values in USD for the social cost of GHGs were adopted from the Interagency Working Group on Social Cost of Greenhouse Gases, United States Government [33]. The social cost values were applied to net emissions for US-Ho1 and for soil chamber measurements to introduce a new GHG social cost forest (GHG-SCF) product that integrates the three gases into a single value of merit for holistic forest management of global warming.

### 2.6. Howland Eddy Covariance Footprint Data

A composite footprint map was made by overlapping layers in Figure 1. The bottom layer consists of a satellite image showing the complete Howland research area redrawn from https://umaine.edu/howlandforest/about/. Then, the CARB measurements area with its plots were redrawn from [17] and overlapped. The top layers are the footprint monthly climatology maps that are in the dataset S3 downloaded from https://zenodo.org/record/4015350 with their backgrounds removed and centered at each tower location. All the Howland footprints available were used (2013 to 2017 for Ho1, and 2003 to 2008 for Ho2 and Ho3). Tower locations and reference circles were highlighted for comparison.

### 2.7. CARB-CAR Data, Documents and Third-Party Verification Review

CARB-CAR Forest methods exclude $CO_2$ measurement relying upon forest mensuration and growth models operationalized over a mandated 100-year project monitoring interval as employed by the California Air Resources Board and Climate Action Reserve [18–20,51]. Howland Forest protocol data for CAR 681 and CAR 1168 results and third-party verification documentation were obtained from the Climate Action Reserve (https://www.climateactionreserve.org/, accessed on 16 February 2021) and the California Air Resources Board (https://ww2.arb.ca.gov/our-work/programs/compliance-offset-program, accessed on 16 February 2021) websites and documents available therein. Supplement Tables S1–S5 provide links to project data and document repositories, cumulative carbon credit performance reports with serial numbers, and historical summary of the CARB-CAR carbon offset supply chain for CAR 681 and CAR1161 and advances in Howland Forest carbon research. Regulations for satisfying AB32 compliance criteria were based on the California Code of Regulations, Title 17, Division 3, Chapter 1, Subchapter 10, Article 5, Sub article 14, Section 95,977(d). Additional information on the CARB mandatory verification process can be found here: https://ww2.arb.ca.gov/our-work/programs/compliance-offset-program/offset-verification, accessed on 16 February 2021.

## 3. Results

The results are presented in Figures 2–5 with supplemental material as indicated. Figure 2 shows the complete GHG record for Howland Forest tower sites (US-Ho1,2,3) expressed in units of $gC\ m^{-2}\ y^{-1}$ and as $tCO_2\ acre^{-1}\ y^{-1}$. Errors are graphed based on CARB-CAR relative to NEE results. Figure 3 provides the underlying annual $R_{eco}$, and GPP data plotted to illustrate the large variance of the $R_{eco}/GPP$ ratio across sites and years. Figure 4 shows the Howland GHG record, including soil chamber data for $CH_4$ and $N_2O$. Figure 5 monetizes the data in Figure 4 based on data provided for each gas's reported social cost for a given year and discount rate.

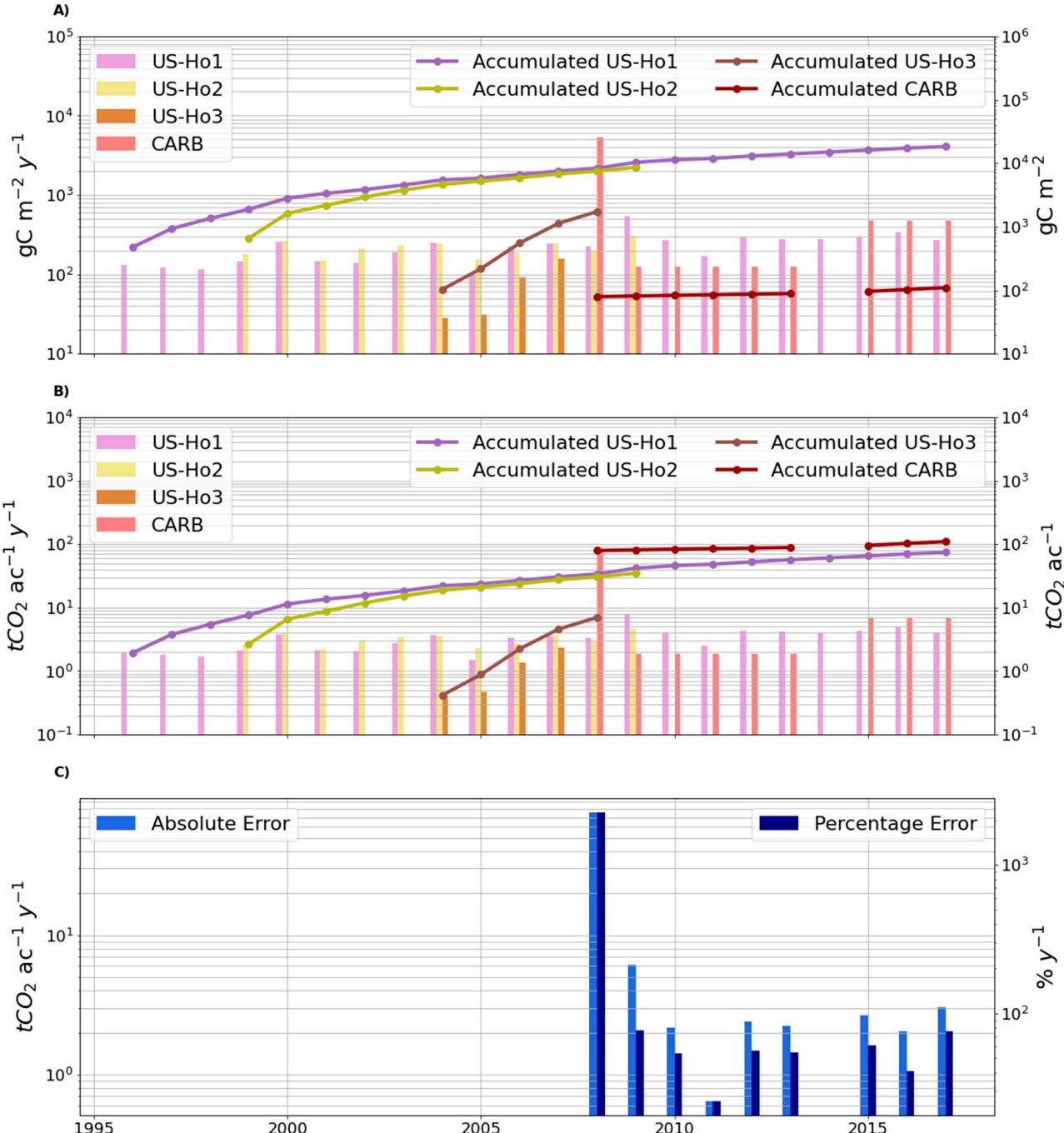

**Figure 2.** Yearly net ecosystem exchange (NEE) budgets (bars) and accumulated values (lines) for US-Ho1 (purple), US-Ho2 (green), US-Ho3 (brown) towers and for CARB (red) data in log-scale and in (**A**) grams of carbon per square meter and (**B**) tons of $CO_2$ per acre. (**C**) Absolute and percentage CARB yearly errors based on direct measurements from US-Ho1, in log-scale. Negative error values were made positive to include them in the log scale. A table of errors is provided in the Supplement Table S6.

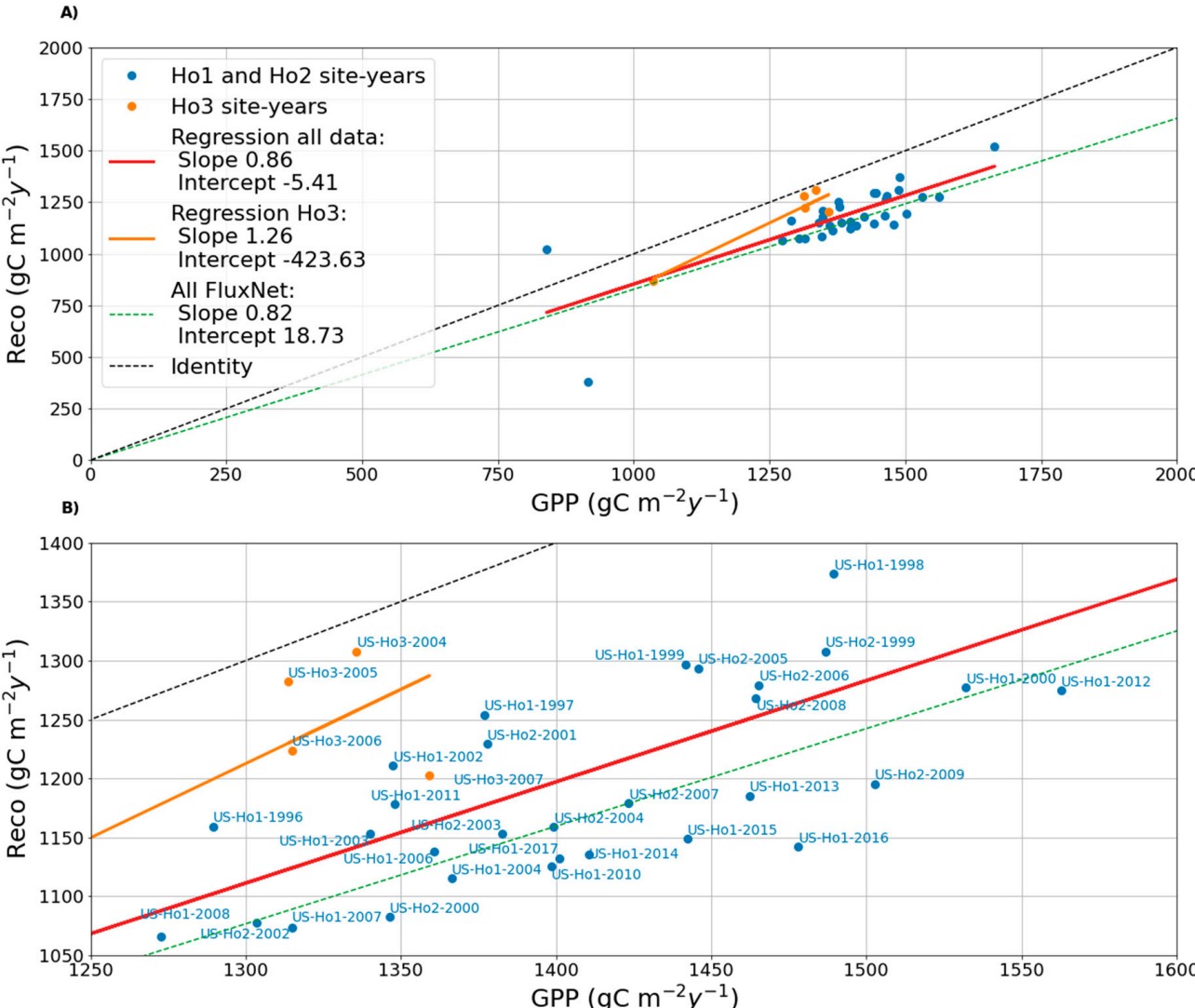

**Figure 3.** Yearly R$_{eco}$ vs. gross primary productivity (GPP) for US-Ho1, US-Ho2 (blue points) and US-Ho3 (orange points). The regression lines (using all the data, solid red line; using only US-Ho3 data, solid orange line). The identity (dashed black line) and FLUXNET regression (dashed green line) were added as a reference. (**A**) All the years; (**B**) only years in the ranges (1050; 1400) and (1250; 1600) gC m$^{-2}$ y$^{-1}$ for R$_{eco}$ and GPP, respectively.

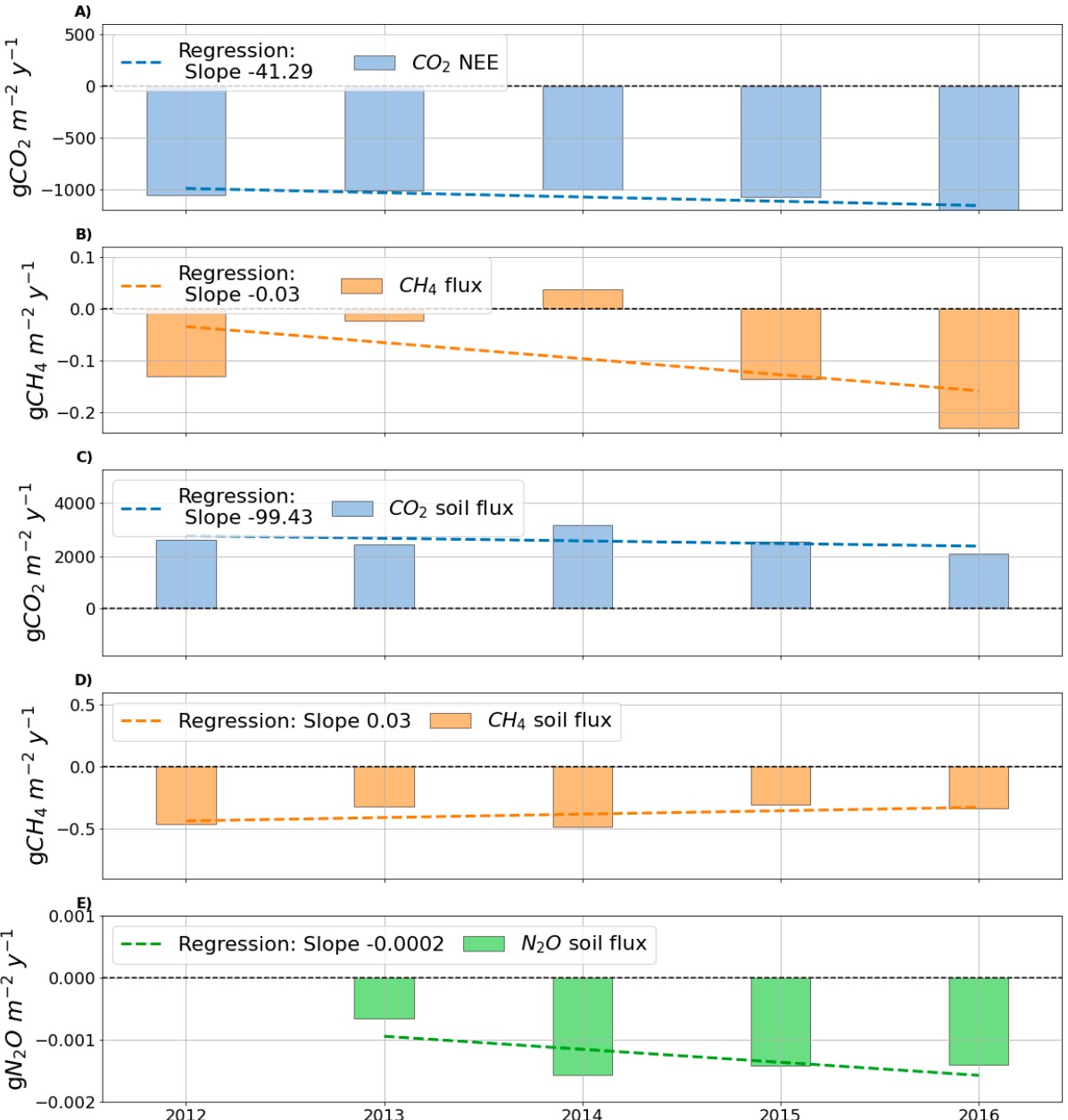

**Figure 4.** Yearly budgets (bars) and trends (lines) for $CO_2$ (**A**) and $CH_4$ (**B**) fluxes measured at the top of the tower, and soil $CO_2$ (**C**), $CH_4$ (**D**) and $N_2O$ (**E**) fluxes measured at chambers no. 2 ($CO_2$ and $CH_4$) and no. 10 ($N_2O$). Soil fluxes budgets from (**C–E**) correspond only to the period May–November, while the tower fluxes from (**A,B**) to the full year [38].

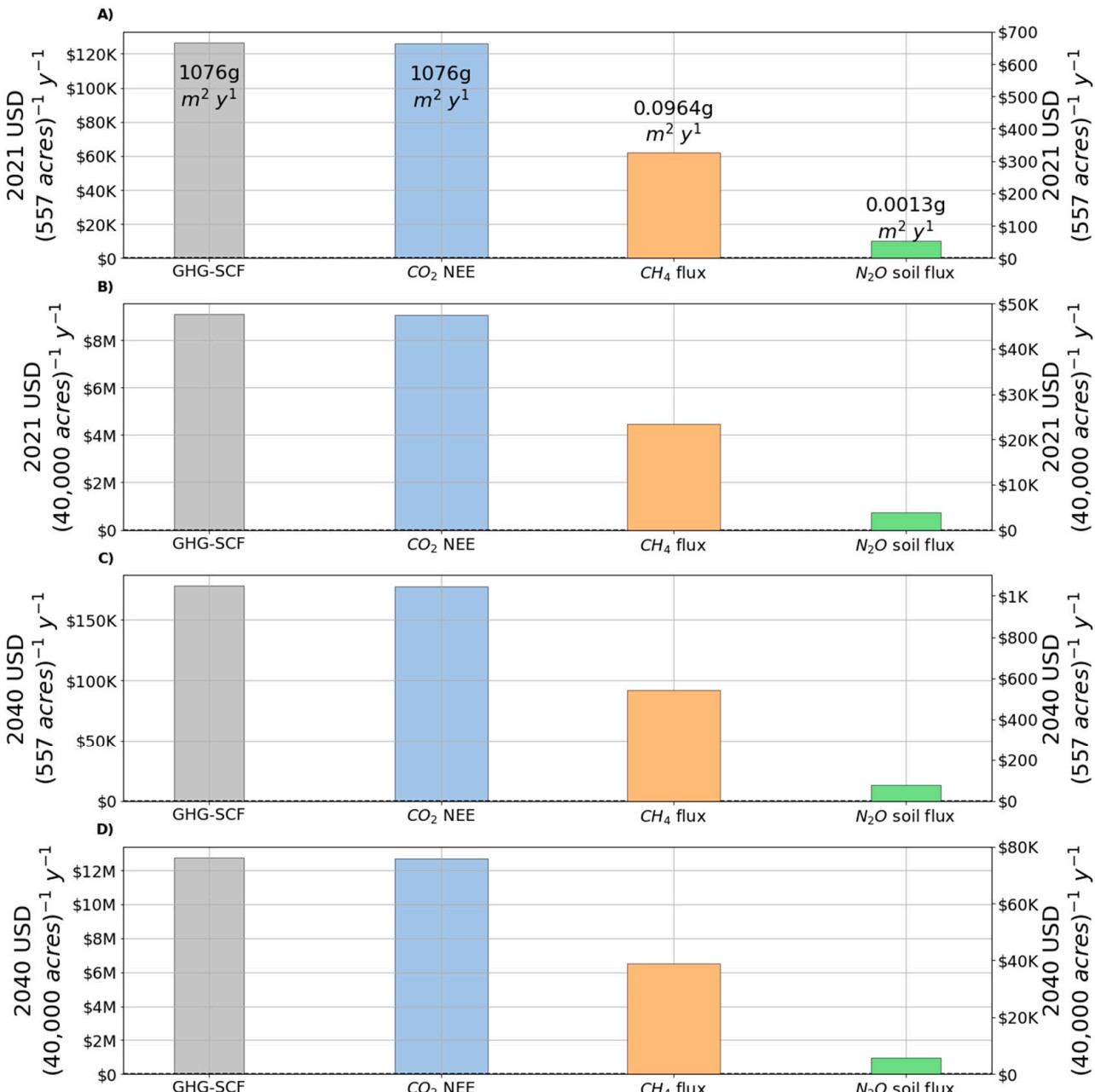

**Figure 5.** Greenhouse gas (GHG) social cost forest (GHG-SCF) yearly means (bars) for $CO_2$ (blue) and $CH_4$ (orange) fluxes measured at the top of the tower, $N_2O$ (green) fluxes measured at chambers no. 2 ($CO_2$ and $CH_4$) and no. 10 ($N_2O$), and the total sum of the three (GWG economic value; gray). Soil flux averages correspond only to the period May–November, while the tower means to the full year. For illustration purposes, results were extrapolated to the Howland research area (557 acres; panels (**A,C**) and to 40,000 acres (panels (**B,D**). Social cost values (USD) used were (**A,B**) $52, $1500, and $19,000 per GHG metric ton of $CO_2$, $CH_4$, and $N_2O$, respectively (values estimated for the year 2021). (**C,D**) $73, $2500, and $28,000 per GHG metric ton of $CO_2$, $CH_4$, and $N_2O$, respectively (values estimated for the year 2040) [33]. In each panel, GHG-SCF Economic values and $CO_2$ NEE are represented by left Y-axes, while $CH_4$ and $N_2O$ fluxes by right Y-axes. In panel (**A**), yearly means capture is annotated for each gas in g m$^{-2}$ y$^{-1}$ above each bar.

### 3.1. GHG Record for Howland Forest Tower (US-Ho1) and CARB-CAR (CAR 681, CAR1161)

Figure 2 shows NEE results for US-Ho1, 2, 3, and CARB-CAR values for the combined period of 1996 to 2017; absolute error results for the period 2008 to 2017 are shown (see also Supplement Table S6 for corresponding percentage error data). (A) NEE gC m$^{-2}$ y$^{-1}$,

B) NEE t$CO_2$ acres y$^{-1}$ (ac$^{-1}$ y$^{-1}$); shown (A,B) are annual and cumulative values (years 2008 to 2013 represent CAR 681, years 2015 to 2017 represent CAR1161), (C) absolute and percent errors for CAR annual project data relative to NEE overlapping data. Net carbon sequestration values are shown as positive. The mean and standard deviation for the aggregated CAR time-series (2008–2013, 2015–2017) of −768.1 ± 1716 gC m$^{-2}$ y$^{-1}$ is ~2.7x the mean of −288.3 and ~17x the standard deviation of 95.5 gC m$^{-2}$ y$^{-1}$, for Howland NEE Ho1 and Ho2 towers, respectively, consistent with exceeding the CARB 5% invalidation threshold [52] and natural ranges for reported interannual NEE variance [1,23]. CAR681 reports a 2008 value of −43,687.0 t$CO_2$e, or −5334.7 gC m$^{-2}$ y$^{-1}$, ~19x larger than the Ho-1 NEE 2008 value of −287.1 gC m$^{-2}$ y$^{-1}$, and ~32x the mean value of global forest NEE [53], establishing the CAR681 2008 value as exceeding the known natural ranges for NEE [23,53]. Likewise, CAR681 reports an exact value of 834 t$CO_2$ (e.g., −198.0 gC m$^{-2}$ y$^{-1}$) for the years 2009 to 2013 in contrast to the absence of repeating values for the 20-year US-Ho1 record, documenting incapability of CAR model annual resolution. The CARB-CAR NEE trend is not observed in the natural variance of Ho1. US-Ho1 data are augmented with US-Ho2 NEE data, approximately 775 m apart, representing ~95% of the project footprint area shared by NEE and CARB-CAR data source areas, as shown in Figure 1. A two-tailed t-test comparing US-Ho1 and US-Ho2 returned a *t*-statistic of −0.22 with an associated p-value of 0.83. Thus the difference is not significant. The overlapping years are from 1999 to 2009 (10 years; at 95%, the approximate *t*-threshold is 2.2), consistent with [41]. US-Ho3 documents the recovery of NEE after a shelterwood harvest. The absolute and percentage error of CARB-CAR data relative to NEE data ranges from 0.65 t$CO_2$ ac$^{-1}$ y$^{-1}$, 25.7% to 75.9 t$CO_2$ ac$^{-1}$ y$^{-1}$, and 2258% for the years 2011 and 2008, respectively, if US-Ho1 yearly values are taken as the representative values. CARB-CAR protocol requires a forest inventory on a 12-year basis, reflecting the reliance on annual model simulations in lieu of empirical data (Project Design Document, page 22, Supplement Table S5, Doc # 3).

### 3.2. $R_{eco}$ vs. GPP Plot for Howland Forest $CO_2$ NEE (US-Ho1,2,3)

Figure 3 shows yearly $R_{eco}$ vs. GPP for the three Howland towers (US-Ho1,2 blue points; US-Ho3 orange points) and regression lines (using all data, solid red line; using only US-Ho3 data, solid orange line). The identity (dashed black line) and FLUXNET regression (dashed green line) lines were added as a reference. (A) Shows all the available years, while (B) shows only those in the ranges (1050; 1400) and (1250; 1600) gC m$^{-2}$ y$^{-1}$ for $R_{eco}$ and GPP, respectively. Individual years are identified in B); years with higher GPP relative to $R_{eco}$ likely store more carbon relative to annual records for carbon sequestration shown; larger GPP is offset by larger $R_{eco}$. Outliers in 3-A for US-Ho1 and US-Ho2 (filled blue circles) from lowest to highest GPP are US-Ho1 (2015), US-Ho1 (2009), US-Ho3 (2008), and US-Ho1 (2001).

### 3.3. Annual Budgets of $CO_2$, $CH_4$ and $N_2O$ for Howland Tower (US-Ho1) and Soil Accumulation Chambers

Figure 4 shows yearly budgets (bars) and trends (lines) for $CO_2$ (A) and $CH_4$ (B) fluxes measured at the top of the tower, and soil $CO_2$ (C), $CH_4$ (D) and $N_2O$ (E) fluxes measured at chambers no. 2 ($CO_2$ and $CH_4$) and no. 10 ($N_2O$). Soil fluxes budgets from (C,D and E) correspond only to May–November, while the tower fluxes from (A and B) to the full year. The tower data for $CH_4$ (B) trends to a maximum positive value of ~0.045 g$CH_4$ m$^{-2}$ y$^{-1}$ in 2014, followed by net negative fluxes in 2015 and 2016, resulting in net negative flux with $CO_2$ of −10.8 $CO_2$e. Soil chamber data for $CH_4$ and $CO_2$ do not reflect tower data. The $N_2O$ soil chamber data are net negative, resulting in a net positive flux for the soil areas sampled of 25.7 $CO_2$e [38], based on the limited chamber data available. Mean $CO_2$ and $CH_4$ measured at the three towers were −10.76 and −0.03 metric tons per hectare, respectively, while $CO_2$, $CH_4$ and $N_2O$ mean measured at the soil chambers were 25.8, −0.13 and −0.0037 metric tons per hectare, respectively. Note, however, that the Howland forest is consistently a sink for $CH_4$ and $N_2O$ for the limited periods observed.

*3.4. Monetization of Net GHG Data Using Social Cost Factors and Revenue Projections*

Figure 5 shows the monetization of the integrated average annual GHG social cost for forests (GHG-SCF), (gray bar) for the Howland Forest project (~557 acres) based on the measurement of $CO_2$, $CH_4$ and $N_2O$ at the US-Ho1 tower, or from soil chambers, using estimates for the social cost factor (USD) for each gas for a given year (e.g., USD $52, $15,000, $19,000 for 2021, and $73, $2500, and $28,000 for 2040, for $CO_2$, $CH_4$ and $N_2O$, respectively) [33]. For illustration purposes only, the US-Ho1 measurements were extrapolated to 40,000 acres to demonstrate the potential revenue over larger forest areas. Projections account for social value estimates in 2021 and in 2040. A discount rate of 3% was applied to the values for each gas according to [33], yielding projected project average annual GHG-SCF values (USD) of $124,000 (2021 values, 557 acres) up to $12,200,000 (2040 values, 40,000 acres). Although biomass removal from the Howland Forest project areas (US-Ho1,2) was minimal over the interval of GHG flux measurements, shelterwood harvests would likely reflect the loss of canopy and the resulting reduction in GHG-SCF product value, a trend observed in [43].

CARB-CAR Forest Carbon Supply Chain. Supplement Tables S1–S5 and Document S6 describe the carbon credit supply chain for CARB-CAR forest carbon projects. Supplement Table S1 identifies the transition of CAR681, an improved forest management project (IFM), from an early action project to an eligible ARB compliance project CAR1161 or CAFR5161 listed on 26 February 2015; total offset credits registered for both project numbers are available through the links provided. Supplement Tables S2 and S3 identify the date of issue for specific vintage years and serial numbers for CAR681 and CAR1161, respectively. Supplement Table S4 lists the project documents for CAR681 and CAR1161, covering project data reporting, project design, carbon pools used in calculations and verification documents with dates of entry for each into the CARB regulatory registry. A full verification report for CAR681 credits issued was uploaded on 11 March 2015 (Item 1, SCS Global Services), completing the supply chain for CARB markets for the cap-and-trade AB32 legislated system [16]. A full verification statement is not available for CAR1161 offsets. See Supplement Document S6 for additional details of the analyses presented in Supplement Tables S1–S5.

## 4. Discussion

The three-gas flux inventory ($CO_2$, $CH_4$, $N_2O$) for the Howland Forest [38] demonstrates the commercial promise of expanding direct measurement of forest GHGs, an area of research with limited results [27,54]. The Howland forest project provides an example of net GHG emission footprints coupled with external factors, such as the social cost of GHG emissions [33,55,56], across select areas of the Howland site, resulting in a single value of merit for holistic forest management of global warming. The Howland Forest was a net sink for $CO_2$ and $CH_4$, except for 2014 during the 2012 to 2016 interval for US-Ho1 Figure 4A). Soil accumulation chamber data also consistently demonstrated a sink for $CH_4$ and $N_2O$ but a source for $CO_2$ (Figure 4A). While $CH_4$ and $N_2O$ emissions are 11,200 (Figure 4A) and 828,000 (Figure 4B) orders of magnitude lower than corresponding $CO_2$ fluxes, respectively, they have higher social cost factors relative to $CO_2$ (USD $51) of $1500 (28–36x $CO_2$) and $18,000 (265–298x)x $CO_2$), respectively, calculated for the year 2021 with a 3% discount rate [33]. Projected GHG social cost forest (GHG-SCF) offset products for the Howland project area of 557 acres and extrapolated, for illustration purposes only, to 40,000 acres for 2021 and 2040. Figure 5A–D ranges from (USD) $12,000 (2021, 557 acres) to $12,000,000 (2040, 40,000 acres) (Figure 5A–D). Thus, the contribution of small forest fluxes of non-$CO_2$ GHGs can result in comparatively large revenue benefits that should not be ignored [38] in forest management programs.

The GHG-SCF, as employed in this study, is intended to reflect the societal value of reducing emissions of GHG forest species by one metric ton per year. In principle, the GHG-SCF product includes the value of all climate change impacts, including (but not limited to) changes in human health effects, net agricultural productivity, property damage

from increased flood risk and natural disasters, risk of conflict, environmental migration, and the value of ecosystem services, including those provided by forests [33]. Presently, the variables and mechanisms, such as soil composition, site land-use history, species and age of trees, seasonality, rainfall, and topography, regulating forest GHG gas exchange are not well understood, emphasizing the importance of expanded monitoring of diverse forests [29,54,57]. Direct measurement of GHG-SCF should be an integral part of the realization of green policies (e.g., Green New Deal), providing links to established policy criteria to reduce GHG emissions [33] employing nature-based solutions [12].

The importance of forest carbon respiration to validate net carbon sequestration for Howland is emphasized in Figure 3A,B, showing annual steps in $R_{eco}$ relative to GPP. Figure 3A,B demonstrates that for every annual interval of photosynthetic uptake of $CO_2$ (GPP), there is an obligatory response embodied in $R_{eco}$ [53] or automatic debit to stored carbon intended for carbon trading markets. US-Ho1 $R_{eco}$ vs. GPP for 2008, the initial year of CAR681, yielding a total of 43,687 carbon credits ($-5334.7$ gC m$^{-2}$ y$^{-1}$) (Supplement Table S2), falls within the lower left quadrant of the FLUXNET slope for $R_{eco}$ and GPP values, Figure 3B, lower than most annual intervals. The CAR681 2008 value would require $R_{eco}$ and GPP values of 2667, in the case of the identity line relationship (e.g., $R_{eco}$ = GPP; NEE = 0), clearly outside of the Ho1-3 values; requiring higher values for GPP if considering the regression line for all Ho1-3 data (red regression line). US-Ho1, 2016, showed the highest GPP relative to $R_{eco}$, while US-Ho3 values for 2004–2006 exhibit high respiration relative to GPP. The US-Ho1,2 outliers (Figure 3A) emphasize the need for high-frequency monitoring as anomalous years can have disruptive impacts on project revenue [32]. US-Ho3 confirms the sensitivity of eddy covariance NEE to timber harvest and regrowth (Figure 2A,B), a trend not detected by CARB-CAR methods but a requirement to test CARB-CAR modeled harvest and growth simulations [17]. Eddy covariance data provide insights into carbon dynamics and related economics not possible with biometric surveys conducted every 6- to 12-years, typical for the Howland CARB-CAR protocol [17].

Considering $CO_2$ alone, Howland Forest NEE tower data, US-Ho2, in conjunction with US-Ho1, covers ~95% of the shared project footprint area with CARB-CAR forest plots (Figure 1). NEE values for US-Ho1 and US-Ho2 are comparable, lacking significant differences between the towers. The Howland two-tower NEE data confirms irreconcilable differences for carbon accounting relative to CARB-CAR data consistent with previous results [1] of offset over-crediting and overpayment by ~4x relative to NEE values [1]. The aggregate CAR681 and CAR1161 time series (2008–2103, 2015–2017) was ~2.7x the mean and ~17x the standard deviation for Howland NEE over the same period, exceeding the 5% invalidation threshold cited by CARB [52] and lying outside of the natural range for 20 years of measured interannual Howland [58] and NEE forest values [23,53]. The exclusion of ecosystem respiration terms for $CO_2$ within the CARB-CAR protocols, critical for calculation of net forest carbon sequestration, confirm incomplete carbon accounting and likely erroneous, invalid offsets for the CARB compliance process for CAR681 and CAR1161. Absent ecosystem respiration, errors of up to 2258% per year were calculated (Figure 2C; Supplement Table S6), emphasizing the requirement for complete carbon accounting, consistent with the well-characterized relationship between $R_{eco}$ and GPP (Figure 2A,B), and soil chamber measurements for $CO_2$ efflux (Figure 5).

The three-gas forest eddy-covariance systems employed at Howland are comprised of commercially available single and multi-gas analyzers for eddy covariance (e.g., $CO_2$ and $CH_4$, $N_2O$) [38,58–60], also applicable to soil chamber gas analyses [38]. A combination of three-gas eddy covariance tower networks of varying heights and soil chamber measurement campaigns can be scaled up across specific ecosystem landscapes by employing expanded ground networks, increasingly inclusive of $CH_4$ and $N_2O$ monitoring [61–64], scale-aware models [65], and remote sensing data [66,67] available for the US and increasingly across the planet [68]. In contrast to the diversity of GHG direct measurements and applications for Howland, the CARB-CAR protocols identify and list $CH_4$ and $N_2O$ only as sources [20], precluding determination of net budgets for these gases, and emphasizing

limitations of estimation protocols that exclude direct measurement of GHGs. Equations for net GHG reductions and removal enhancements cited in [20] may apply to any GHG but are uniquely denominated for $CO_2$, as source or sink, linked to carbon and tree growth equations and models to satisfy the 100-year carbon baseline and tree harvest scenarios required for CARB-CAR products. Accordingly, CARB-CAR protocol uncertainties, if employed to determine offsets for non-$CO_2$ GHGs, are likely higher than for $CO_2$ and limited to source emissions rather than net emissions for these gases.

Independent verification of emission reduction claims is critical to the integrity of GHG offset supply chains. Analysis of third-party verification of the CARB-CAR forest carbon supply chain revealed inconsistencies with CARB-CAR policy, including: (1) The CARB-CAR Howland project did not meet CARB reporting regulations for both tranches of Howland CARB offsets as an early action project (CAR681), or as an ARB compliance project (CAR1661), by noncompliance of offset verification reporting dates (Supplement Table S5, Supplement Document S6); (2) CAR misstated the actual values for a single year of NEE data (1996) [41] as seven years of seasonal Howland NEE data in support of CAR model adjustment for seasonal trends in tree growth. However, the CAR model (v. 3.2) excludes terms for soil carbon as ecosystem respiration, intrinsic to NEE data, and requires conversion of NEE micromoles $m^{-2}\,s^{-1}$ to tree volume, a complex topic addressed by [69]. Details of model revisions and results were not provided, calling the validity of model results into question; (3) The Howland NEE and soil GHG records, advancing annually from 1996 to 2021, were available to CARB-CAR project owners, operators, and third-party verifiers (38,40,41,43,44,45,58,59,60) overlapping with the supply chain process from 2013 to 2019 culminating in serialized CARB verified offsets according to the AB32 mandate [70]. The Howland US-Ho1 NEE data were not reported as an independent check of the CARB-CAR annual results, a comparison that would have constrained the natural ranges for carbon sequestration offering an opportunity to proscriptively avoid CARB-CAR forest carbon sequestration uncertainties; (4) The Howland CARB-CAR project reporting exhibits errors and lapses in recordation, similar to those reported previously [1], including numerical errors, changes in reporting format from annual to discretionary mixed time intervals, and non-standard model operations resulting in uncertain values; and, (5) The raw data and detailed model outputs for the CARB-CAR projects have not been made available to the public, limiting collaboration and external verification of the project results. Instead, the CARB-CAR data and information are housed on personal computers with no central repository (Supplement Table S5, Item 10). Considering the uncertainties identified above, the CARB-CAR verification process is scientifically unjustifiable, creating avoidable offset invalidation risk for CAR681 and CAR1161. The exclusion of direct measurement protocols for forest carbon has been recently extended within the Assembly Bill AB398 [71] the successor bill to AB32 [70], by recommendation of a mandated Task Force to guide inclusion of new offset protocols. However, direct measurement of forest carbon protocols was not addressed [72].

The CARB-CAR and similar protocols could be improved by defining measurement and model results within Equation (1) universal reference framework (NEE = $R_{eco}$ + GPP) and incorporating independent field data for direct measurement of $CO_2$. Collaboration with forest carbon sequestration field sites represented by the National Ecological Observatory and the AmeriFlux network of eddy covariance towers [73,74] may suggest improvements in the CARB-CAR protocol. Given the sources of uncertainty identified for the CARB-CAR verification process, improvements could be implemented in the near term, such as providing raw data availability for external users, inter-comparison of CARB-CAR with NEE data where possible, enforcing accounting standards, and adherence to consistent reporting formats.

## 5. Conclusions

The societal value of forests as GHG sinks can be linked with the social cost values established for GHGs emissions. Combining direct measurement of GHGs with their social

cost factors creates a unique GHG social cost forest product (GHG-SCF). The GHG-SCF products represent a single figure of merit for holistic forest management of global warming. Expanding forest measurement of non-$CO_2$ GHGs underscores the limited and uncertain status of typical forest GHG measurement protocols and the importance of independent verification of emission reduction claims.

**Supplementary Materials:** The following are available online at https://www.mdpi.com/article/10.3390/land10040436/s1. The Supplement includes: Tables S1–S6 and Document S6. The following materials are included: Table S1. CAR681 and CAR1161 Project Links. Table S2. CAR681 Project Emission Reductions and Issued Offset Serial Numbers. Table S3. CAR1161 Project Emission Reductions and Issued Offset Serial Numbers. Table S4. CAR681 and CAR1161 Online Project Documents. Table S5. Summary of Howland Timeline for CAR681, CAR1161 and publication of US-Ho1,2,3 data based on Tables S2–S4. Table S6. Absolute and percentage CARB yearly errors if the measurements from US-Ho1 were correct. Positive errors indicate CARB overestimations, while negative values are underestimations. Table S6. Absolute and percentage CARB yearly errors if the measurements from US-Ho1 were correct. Positive errors indicate CARB overestimations, while negative values are underestimations. Document S6. Table S5 CARB-CAR, NEE timeline, and documentation of critical errors.

**Author Contributions:** B.D.V.M. was responsible for the conceptualization, methodology, and writing of the manuscript. N.B. was responsible for analysis, visualization of data and manuscript review. B.R. was responsible for investigation and manuscript review. All authors have read and agreed to the published version of the manuscript.

**Funding:** This research received no external funding.

**Data Availability Statement:** The data used in the analyses presented can be downloaded from the following sources: Howland Forest Tower Data: AmeriFlux US-Ho1 Howland Forest (main tower), dataset: https://doi.org/10.17190/AMF/1246061 (accessed on 17 March 2021). David Hollinger (1999-) AmeriFlux US-Ho2 Howland Forest (west tower), dataset: https://doi.org/10.17190/AMF/1246062. David Hollinger (2000-) AmeriFlux US-Ho3 Howland Forest (harvest site), dataset: https://doi.org/10.17190/AMF/1246063. Howland Ameriflux Long-Term, dataset: https://ameriflux.lbl.gov/sites/site-search/#keyword=Howland. Howland Forest Soil Chamber Dataset: https://doi.org/10.6084/m9.figshare.7445657.v1. Monthly Howland US-Ho1,2,3 Climatology, Dataset: https://zenodo.org/record/4015350.

**Acknowledgments:** The authors acknowledge data availability from FLUXNET2015 (https://fluxnet.fluxdata.org/data/fluxnet2015-dataset) under Tier One data following the guidelines of the CC-BY-4.0 data usage license (Attribution 4.0 International (CC BY 4.0); https://creativecommons.org/licenses/by/4.0/). That license specifies that the data user is free to Share (copy and redistribute the material in any medium or format) and/or Adapt (remix, transform, and build upon the material) for any purpose. https://fluxnet.fluxdata.org/data/data-policy/.

**Conflicts of Interest:** Planetary Emissions Management Inc. is a private research and development organization. No conflicts of interest are declared.

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
