# Peer review of "Howland Forest, ME, USA: Multi-Gas Flux (CO2, CH4, N2O) Social Cost Product Underscores Limited Carbon Proxies"

_land, doi:10.3390/land10040436_

Round 1

Reviewer 1 Report

Peer review of Land ms 1163791
Title: Howland Forest, ME, USA: Multi-gas flux record (CO2, CH4, N2O) establishes new forest products linked to social cost of emission in contrast to carbon limited sequestration proxies,

This manuscript reports results of a long-term study of measurement of green house gases from an installation of eddy covariance towers in Maine for purposes of verification of forest carbon sequestration. 

The information presented is not the subject area of my research investigations. However, I am generally aware of the science of carbon sequestration and particularly the need for its accurate measurement and verification of results.  I reviewed the manuscript from the view of a manager or organization administrators desiring to become more informed on methods for measurement of greenhouse gasses and verification of predicted offset values, the value of those forest products to society and possible magnitudes of carbon accounting errors. I consider important the information presented on annual revenue for greenhouse gas products sold as a forest offset product and annual revenues for those products. This study demonstrates the possible magnitude of sequestration errors resulting from incomplete carbon accounting.

Although the manuscript reads somewhat like an office final report, I have no specific suggestions for revising text where the manuscript could be modified for better presentation of the study findings. 

Author Response

AUTHORS RESPONSE TO REVIEWER 1 ARE MARKED IN RED.

1. Thank you for taking the time to review the manuscript and provide your summary of the findings from an administrator or policy advisor's perspective. 

Peer review of Land ms 1163791
Title: Howland Forest, ME, USA: Multi-gas flux record (CO2, CH4, N2O) establishes new forest products linked to social cost of emission in contrast to carbon limited sequestration proxies,

This manuscript reports results of a long-term study of measurement of green house gases from an installation of eddy covariance towers in Maine for purposes of verification of forest carbon sequestration. 

The information presented is not the subject area of my research investigations. However, I am generally aware of the science of carbon sequestration and particularly the need for its accurate measurement and verification of results.  I reviewed the manuscript from the view of a manager or organization administrators desiring to become more informed on methods for measurement of geenhouse gasses and verification of predicted offset values, the value of those forest products to society and possible magnitudes of carbon accounting errors.

I consider important the information presented on annual revenue for greenhouse gas products sold as a forest offset product and annual revenues for those products. This study demonstrates the possible magnitude of sequestration errors resulting from incomplete carbon accounting.

2. Identifying the economic value of nature-based solutions to climate by non-scientists is the ultimate goal, in my opinion, of publishing in this field. Likewise, an awareness of uncertainties in methods of carbon accounting is essential to evaluate the use of methods. 

Although the manuscript reads somewhat like an office final report, I have no specific suggestions for revising text where the manuscript could be modified for better presentation of the study findings. 

Thank you.

Reviewer 2 Report

This paper touches upon an important issue within the debate on the concept of Multi-gas flux and the social cost of emission scheme.  This paper show an innovative method (Eddy covariance method) that analyzes the gas flux. This paper presents also the social cost of emission and all the consequence around it. The framework and the theoretical scheme are quite interesting but not really clear.

Before providing detailed comments to the specific sections, I have some general suggestions to strengthen the analytical consistency.

Overall comment

The authors need to reframe the result and conclusion section. If the introduction very good, it is still lacking the conceptual framework. The method section can be reduced

Your paper is not a report, it is presented now as a report.  Subsection and good title have to be design to address all the research questions.

Conclusion is still thin, more development can be done.

Authors can develop more the conclusion

Having said that, I have some specific comments and suggestions to improve the manuscript.

 Title should be review, should be reduced

Line 12-13: Yes, but not only.

Keywords: Reduce keywords to five

apart from SOFO 2020, you can look http://www.fao.org/forest-resources-assessment/2020/en/

line 43: 6-12 years only?? which inventory method? Typology of the forest? where

line 46-50: this sentence is too long, rephrase it

2.1. site description

Figure 1: Forest location is not clear. Where are the boundaries? A separate map should be produce...

Line 87--93, you can develop more the key on the map

Also name the towers, to differentiate them in this map

2.2. CO2...

Line 113-116: add this mail in the figure 1

2.3. CO2, CH4....

2.4. data...

All the result section: good result but this section is not design in the good way to do it.  Author should disaggregate the result section into subsection with a good title.

figure 4: this is not the good way to present  the result.

table 1 to table 5: al this should be shift to  the Appendix

line 353: relativizes all this

the conclusion should be more developed

All remarks and comments are in the manuscript.

Hope these comments are helpful to improve the manuscript for submission in LAND .

Author Response

This paper touches upon an important issue within the debate on the concept of Multi-gas flux and the social cost of emission scheme.  This paper show an innovative method (Eddy covariance method) that analyzes the gas flux. This paper presents also the social cost of emission and all the consequence around it. The framework and the theoretical scheme are quite interesting but not really clear.

Before providing detailed comments to the specific sections, I have some general suggestions to strengthen the analytical consistency.

Overall comment

The authors need to reframe the result and conclusion section. If the introduction very good, it is still lacking the conceptual framework. The method section can be reduced

Your paper is not a report, it is presented now as a report.  Subsection and good title have to be design to address all the research questions.

Conclusion is still thin, more development can be done.

Authors can develop more the conclusion

Having said that, I have some specific comments and suggestions to improve the manuscript.

 Title should be review, should be reduced

Line 12-13: Yes, but not only.

Keywords: Reduce keywords to five

apart from SOFO 2020, you can look http://www.fao.org/forest-resources-assessment/2020/en/

line 43: 6-12 years only?? which inventory method? Typology of the forest? where

line 46-50: this sentence is too long, rephrase it

2.1. site description

Figure 1: Forest location is not clear. Where are the boundaries? A separate map should be produce...

Line 87--93, you can develop more the key on the map

Also name the towers, to differentiate them in this map

2.2. CO2...

Line 113-116: add this mail in the figure 1

2.3. CO2, CH4....

2.4. data...

All the result section: good result but this section is not design in the good way to do it.  Author should disaggregate the result section into subsection with a good title.

figure 4: this is not the good way to present  the result.

table 1 to table 5: al this should be shift to  the Appendix

line 353: relativizes all this

the conclusion should be more developed

Reviewer 3 Report

Reduce the keywords to 5 to 7

Author Response

Reduce the keywords to 5 to 7

-------------------------------------------------

AUTHOR RESPONSES ARE PRESENTED IN RED

Reduce the keywords to 5 to 7

  • Keywords reduced to five.
  • Revised Lines 31 – 32: Keywords: California Air Resources Board; Climate Action Reserve; eddy covariance; Howland Forest; social cost of CO2, CH4, N2O
  • Thank you for your suggestions.

Reviewer 4 Report

This paper deal with the problem of dealing with non-CO2 greenhouse gases in the computation of total net sequestration in forest ecosystems. The authors develop a complex procedure that includes specific tower fluxes and eddy covariance methods for data processing. The paper has two interesting new contributions. One is that it integrates these gases from a physical and economic point of view. The other contribution is the extrapolation of the results to a more extensive area, and a longer time series. The shortcomings of the paper are: mainly, it is difficult to follow; in particular, the explanation of all the project documents is very complicated. The authors should try to focus on their measurements and results and moving those Tables to an Annex (Tables 2 to 5). The presentation of results also needs more development. Another point, make more explicit the contribution of the paper. I think the authors could justify more clearly the advantages of incorporating non-CO2 greenhouse gases and their transaction costs, if they exist. More detailed comments below:

•    I think a clear explanation of the contribution of the paper could be developed. 
•    The concept of social cost needs more explanation. Besides, it is not clear how the economic analysis has been made. Is it only an NPV with a 3% discount rate? Does it not include the costs to obtain measures of the non-CO2 greenhouse gases? Besides, I do not understand the figures in Figure 5(52, 1500, and 19000 USD dollars) Where do they come from?
•    It is hard to understand some issues regarding the errors (l. 324-325). Maybe a new Table could be helpful.
•    What happens if in the case study appear thinnings, final cut, etc.?
•    The Conclusions section should be rewritten.

Author Response

This paper deal with the problem of dealing with non-CO2 greenhouse gases in the computation of total net sequestration in forest ecosystems. The authors develop a complex procedure that includes specific tower fluxes and eddy covariance methods for data processing. The paper has two interesting new contributions. One is that it integrates these gases from a physical and economic point of view. The other contribution is the extrapolation of the results to a more extensive area, and a longer time series. The shortcomings of the paper are: mainly, it is difficult to follow; in particular, the explanation of all the project documents is very complicated. The authors should try to focus on their measurements and results and moving those Tables to an Annex (Tables 2 to 5). The presentation of results also needs more development. Another point, make more explicit the contribution of the paper. I think the authors could justify more clearly the advantages of incorporating non-CO2 greenhouse gases and their transaction costs, if they exist. More detailed comments below:

•    I think a clear explanation of the contribution of the paper could be developed. 
•    The concept of social cost needs more explanation. Besides, it is not clear how the economic analysis has been made. Is it only an NPV with a 3% discount rate? Does it not include the costs to obtain measures of the non-CO2 greenhouse gases? Besides, I do not understand the figures in Figure 5(52, 1500, and 19000 USD dollars) Where do they come from?
•    It is hard to understand some issues regarding the errors (l. 324-325). Maybe a new Table could be helpful.
•    What happens if in the case study appear thinnings, final cut, etc.?
•    The Conclusions section should be rewritten.

Round 2

Reviewer 4 Report

Dear Authors,

Good job! After reading the new version of the manuscript and the authors' responses, I believe the manuscript has been significantly improved, and it can now be published in Land.

Best regards